# The Impact of Novel Reconstruction Algorithms on Calcium Scoring: Results on a Dedicated Cardiac CT Scanner

**DOI:** 10.3390/diagnostics13040789

**Published:** 2023-02-20

**Authors:** Milán Vecsey-Nagy, Zsófia Jokkel, Ádám Levente Jermendy, Martin Nagy, Melinda Boussoussou, Borbála Vattay, Márton Kolossváry, Csaba Csobay-Novák, Sigal Amin-Spector, Béla Merkely, Bálint Szilveszter

**Affiliations:** 1Heart and Vascular Center, Semmelweis University, 68. Varosmajor Street, 1122 Budapest, Hungary; 2Gottsegen National Cardiovascular Center, 29. Haller Street, 1096 Budapest, Hungary; 3Physiological Controls Research Center, University Research and Innovation Center, Óbuda University, 96/b Bécsi út, 1034 Budapest, Hungary; 4Arineta Ltd., 15. Khalamish Street, P.O. Box 3057, Caesarea 3088900, Israel

**Keywords:** coronary artery disease, coronary artery calcium score, image reconstruction, cardiovascular risk

## Abstract

Contemporary reconstruction algorithms yield the potential of reducing radiation exposure by denoising coronary computed tomography angiography (CCTA) datasets. We aimed to assess the reliability of coronary artery calcium score (CACS) measurements with an advanced adaptive statistical iterative reconstruction (ASIR-CV) and model-based adaptive filter (MBAF2) designed for a dedicated cardiac CT scanner by comparing them to the gold-standard filtered back projection (FBP) calculations. We analyzed non-contrast coronary CT images of 404 consecutive patients undergoing clinically indicated CCTA. CACS and total calcium volume were quantified and compared on three reconstructions (FBP, ASIR-CV, and MBAF2+ASIR-CV). Patients were classified into risk categories based on CACS and the rate of reclassification was assessed. Patients were categorized into the following groups based on FBP reconstructions: 172 zero CACS, 38 minimal (1–10), 87 mild (11–100), 57 moderate (101–400), and 50 severe (400<). Overall, 19/404 (4.7%) patients were reclassified into a lower-risk group with MBAF2+ASIR-CV, while 8 additional patients (27/404, 6.7%) shifted downward when applying stand-alone ASIR-CV. The total calcium volume with FBP was 7.0 (0.0–133.25) mm^3^, 4.0 (0.0–103.5) mm^3^ using ASIR-CV, and 5.0 (0.0–118.5) mm^3^ with MBAF2+ASIR-CV (all comparisons *p* < 0.001). The concomitant use of ASIR-CV and MBAF2 may allow the reduction of noise levels while maintaining similar CACS values as FBP measurements.

## 1. Introduction

Coronary artery disease (CAD) is responsible for approximately half of cardiovascular disease (CVD) associated deaths; meanwhile, CVDs remain the leading cause of morbidity and mortality worldwide [1,2]. Although there are several pharmacological methods and lifestyle options available for primary and secondary prevention of CVDs, identifying patients who could benefit most from early initiation of therapy remains a significant challenge [3]. Traditional risk models incorporating markers such as age and sex provide limited accuracy at an individual level to guide patient management [4]. Therefore, further methods are needed for the prediction of CV risk in order to provide precise early detection and risk stratification for patients [5].

Coronary artery calcium score (CACS) is a robust non-invasive diagnostic marker for the quantification of calcified coronary atherosclerotic burden [6]. CACS can be measured using non-contrast cardiac CT scans with relatively low radiation exposure; nevertheless, it provides valuable prognostic information regarding future cardiovascular events in both symptomatic and asymptomatic patients [7]. Several population-based studies have analyzed the prognostic value of CACS in the prediction of CAD [8,9]. Given that CAC-based risk assessment relies on a cost-effective and widely available non-invasive imaging modality, several international guidelines incorporated CACS into their models of risk assessment and therapy management [10,11]. Since CCTA has recently received a class I recommendation for the assessment of patients with chronic coronary syndrome, a substantial increase can be projected in the number of scans performed [12]. The prevailing concern of associated ionizing radiation, however, warrants the need for radiation-conscious approaches [13]. One potential means of enabling lower radiation is the utilization of iterative reconstructions (IR) [14,15].

The traditional methodology for CAC scoring developed by Agatston et al. applied a protocol with standard acquisition parameters (120 kVp and 3 mm slice thickness) and image reconstruction (filtered back projection, FBP), which enabled highly reproducible measurements of coronary calcification across different vendors [16]. IR algorithms, on the other hand, may be used in CT to reduce image noise and improve image quality, and as such, they may be exploited for radiation reduction purposes [17,18]. For this purpose, numerous IR techniques were developed for coronary CT scans by vendors as an alternative to traditional FBP [19,20,21,22,23].

The world’s first dedicated cardiac CT scanner (CardioGraphe, GE Healthcare, Chicago, IL, USA) was introduced in 2017 and several vendor-specific reconstructions have been developed to meet the requirements of this purpose-built scanner [24]. The new-generation adaptive statistical IR developed for CV imaging (ASIR-CV) and the model-based adaptive filter (MBAF2) were both designed to optimize image quality and the latter has demonstrated promising results in increasing the contrast-to-noise and signal-to-noise ratio of coronary CT angiography (CCTA) datasets [25]. Proper quantification of coronary artery calcification and the reproducible nature of measurements on non-contrast CT scans is, nevertheless, of utmost importance as inaccurate calculations could lead to false risk classification and potentially misguided clinical management of patients [26].

Our aim was to compare CACS measurements performed with ASIR-CV to FBP calculations and to investigate the potential incremental value of combining it with MBAF2. We also sought to demonstrate possible CV risk reclassification tendencies based on the different CACS results obtained with the different reconstruction techniques.

## 2. Materials and Methods

### 2.1. Participants

In the current retrospective study, we examined 404 consecutive patients who underwent elective CCTA scanning due to stable angina between September 2021 and May 2022. We collected the medical history and the demographic data of our subjects including age, sex, BMI, and cardiovascular risk factors. We excluded patients with previous coronary stent implantation, coronary artery bypass grafts (CABG), or any implanted devices that may cause artifacts at any coronary segments on our scans, such as mechanical aortic valves or pacemaker electrodes. CT scans with significant motion artifacts were also excluded. Due to the retrospective nature of the study, informed consent was waived by the institutional review board.

### 2.2. Coronary CT Scans

Coronary CT scans were obtained at a tertiary referral center using a dedicated cardiac CT scanner (CardioGraphe, GE Healthcare, Chicago, IL, USA). Patients were given oral and/or intravenous beta blockers as needed to reach a target heart rate of 65 beats per minute. We obtained prospective ECG-gated non-contrast scans for calcium scoring with 280 × 0.5 collimation, 240 ms tube rotation time, and 120 kVp tube voltage. The tube current was adjusted according to the size of the patient. We registered the total dose length product (DLP) for each patient and calculated the effective radiation dose by multiplying it by the conversion factor of 0.014 mSv × mGy^−1^ × cm^−1^.

Reconstructions from the raw data were obtained on a dedicated workstation (GE Healthcare, Chicago, IL, USA). The raw dataset of each patient was reconstructed using FBP, ASIR-CV, and the combination of MBAF2 with ASIR-CV.

### 2.3. Quantification of Image Noise

Image noise was determined as the standard deviation of the attenuation value in a single circular 200 mm^2^ region of interest (ROI) placed in the aortic root at the level of the left main coronary ostium. For all measurements, the size, shape, and position of the ROIs were kept constant among the three reconstruction techniques using dedicated open-source software (Osirix, version 12.0; Osirix Foundation, Geneve, Switzerland).

### 2.4. Coronary Calcium Score Calculation

We measured CACS based on the Agatston method on axial series with the dedicated built-in software of Philips IntelliSpace Portal (Heartbeat-CS, Philips Healthcare, Best, Netherlands) [16]. With this semi-automatic method, the software can detect voxels above a density of 130 Hounsfield units and an area of 1 mm^2^. Calcifications were identified on all three reconstructions (FBP, ASIR-CV, and MBAF2+ASIR-CV) by a trained radiologist with 6 years of experience in coronary CT. Patients were classified into the following CV risk categories based on the overall Agatston scores: 0: no identifiable calcification; 1–10: minimal calcification; 11–100: mild calcification; 101–400: moderate calcification; 400<: severe calcification. We analyzed the risk classification of each patient based on the three reconstruction methods and determined the number of patients who needed reclassification to a different CV risk group due to the utilization of ASIR-CV or MBAF2+ASIR-CV. We demonstrate a representative case of reclassification on a 61-year-old patient in Figure 1.

### 2.5. Statistical Analysis

Continuous variables are expressed as mean ± standard deviation (SD) and categorical variables are expressed as numbers and percentages. To evaluate the normality of the continuous parameters, we applied the Kolmogorov–Smirnov test. Image noise parameters, CACS values, and calcium volumes were compared with the Friedman test and post hoc pairwise comparisons have been performed using Wilcoxon’s signed-rank test. In all our calculations, a *p*-value of < 0.05 was considered significant. We performed Bland–Altman analysis and calculated Lin’s concordance correlation coefficients for the differences between the Agatston scores with FBP and ASIR-CV, as well as MBAF2+ASIR-CV. We performed statistical analysis using SPSS (Armonk, NY, USA, version 27.0).

## 3. Results

From a series of 446 consecutive patients who underwent CCTA in the inclusion period, we excluded 34 patients because of prior PCI or CABG, 6 patients due to implanted intracardiac device, and 2 patients because of inadequate CT image quality (Figure 2).

Overall, 404 patients including 146 females participated in our study with a mean age of 57.5 ± 12.2 years and an average body mass index of 27.9 ± 4.6 kg/m^2^. The demographic data and CT characteristics of our study group are summarized in Table 1.

### 3.1. Quantification of Image Noise

As shown in Figure 3, when compared to FBP, ASIR-CV and ASIR-CV+MBAF2 both resulted in reduced image noise (21.5 ± 6.7, 14.3 ± 4.4, and 12.8 ± 4.0 HU, respectively, all pairwise comparisons *p* < 0.001) with ASIR-CV+MBAF2 demonstrating the most marked impact.

### 3.2. Calcium Score Classifications

Based on FBP reconstructions, 172 patients had a CACS of 0, 38 had values between 1 and 10, 87 were classified to the 11–100 category, 57 to the 101–400 category, and 50 to the 400< category. Using the MBAF2 method combined with ASIR-CV, several patients shifted from a higher CV risk group to a lower: 0.5% (2/404) shifted from minimal to zero, 2.5% (10/404) from mild to minimal, 0.7% (3/404) from moderate to mild, and 1.1% (4/404) from severe to moderate. With the stand-alone utilization of ASIR-CV, the same patients were recategorized as in the MBAF2+ASIR-CV method and besides them, a few additional patients shifted from a higher-risk group to a lower category: 1.2% (5/404) shifted from minimal to zero, 3.0% (12/404) from mild to minimal, 1.2% (5/404) from moderate to mild, and 1.2% (5/404) from severe to moderate. For better visualization, we present our results in a Sankey diagram (Figure 4).

### 3.3. Total Calcium Volume and Agatston Score Comparisons

Overall, median calcium volume was 7.0 (0.0–133.25) mm^3^ with the gold-standard FBP reconstruction, 4.0 (0.0–103.5) mm^3^ with ASIR-CV, and 5.0 (0.0–118.5) mm^3^ using MBAF2+ASIR-CV with all pairwise comparisons yielding a *p*-value of < 0.001. Overall, Agatston score was 17.0 (0.0–324.8) with FBP reconstruction, 9.5 (0.0–286.5) with ASIR-CV, and 11.0 (0.0–288.0) with MBAF2+ASIR-CV, all comparisons with a *p*-value of < 0.001. These results are summarized in Table 2.

The concordance correlation coefficient of ASIR-CV with FBP was 0.992 for Ca volume and 0.996 for Agatston score. The concordance correlation coefficient of MBAF2+ASIR-CV with FBP was 0.996 for Ca volume and 0.998 for Agatston score (Table 3).

We demonstrated the differences in Agatston scores between standard FBP and the different reconstruction techniques using Bland–Altman plots (Figure 5). Compared with FBP, ASIR-CV underestimated calcium score by a mean of 14.1; meanwhile, the combination of MBAF2 and ASIR-CV measured 10.5 less than FBP.

## 4. Discussion

The current investigation uniquely demonstrated that contemporary denoising reconstruction algorithms developed for a purpose-built cardiac CT display a tendency of underestimating CACS, as compared to the gold-standard FBP. Both ASIR-CV and the combination of MBAF2 and ASIR-CV reduce CAC volumes and Agatston scores significantly, however, the degree of discrepancies and the rate of downgrade CV risk reclassifications can be mitigated by implementing the combined approach. ASIR-CV and MBAF2 are vendor-specific and relatively new algorithms developed recently for a dedicated cardiac scanner. Although the impact of MBAF2 on CCTA datasets has been studied before, scant evidence is available about its effect on non-enhanced scans. Therefore, this study aids by providing previously unpublished data on this matter and by contributing to the currently available literature with regard to the effect of ASIR-CV and MBAF2 on image quality. Furthermore, to the best of our knowledge, this is the first study to assess the impact of the combination of these two dedicated reconstruction algorithms on the quantification of CAC volume and score.

The incremental value of different vendor-specific iterative reconstructions on image quality has extensively been investigated in previous studies [18,27,28]. In a recent study, however, Lim et al. demonstrated similar beneficial tendencies with a vendor-neutral algorithm as well using an anthropomorphic phantom [29]. These techniques have hence been hypothesized to be a valid resource for more radiation-conscious approaches in clinical imaging. The PROTECTION V trial was a multicenter, multivendor prospective study that assessed the potential feasibility of radiation dose reduction with iterative reconstructions in CCTA. As a conclusion of this trial, they found that CCTA image quality was maintained after concomitantly implementing a 30% reduction in tube current and iterative reconstruction algorithms, as compared to conventional FBP and standard tube current [30]. Similar tendencies can be envisioned for the dedicated ASIR-CV and MBAF2 algorithms as well, and future investigations in this matter could unequivocally prove to be beneficial to a wide array of patients.

CACS has emerged as a reliable early CV risk marker in the past decades [31]. It is a non-invasive and relatively safe method that has recently been added to the recommendations of the European Society of Cardiology for the early detection of CAD in symptomatic and asymptomatic patients with intermediate risk for CV disease [32,33,34]. In accordance with current guidelines, however, an inherent steady increase can be projected in the number of performed cardiac CTs in the future, which emphasizes the importance of dose reduction techniques [35]. In order to meet contemporary clinical needs, vendors are continuously exploring ways to develop more precise and advanced technologies for the post-processing of CCTA datasets to achieve lower radiation exposure while maintaining both consistent calcium quantification and an acceptable level of image noise [36,37]. IRs have emerged as promising techniques in CCTA where the beneficiary impact has extensively been investigated regarding image quality and diagnostic accuracy [14,38]. Nevertheless, given that IRs are expected to decrease CACS due to the inherent coupling of the score and image noise, modern CACS quantification cannot fully utilize the dose mitigation techniques implemented in CCTA investigations. Several previous studies have investigated the effect of dose reduction protocols in combination with IR on CACS values; however, the results were inconclusive. A few studies reported that IR had no significant effect on the Agatston score [39,40]. On the contrary, the majority of the studies reported a significant reduction in CACS with IR which was not limited to one vendor’s IR algorithm; trends were observed in all four major CT vendors’ reconstruction models [25,41,42,43,44]. In accordance with previous evidence, a contemporary IR algorithm (ASIR-CV) significantly decreased CACS and calcium volumes in our current study; however, the degree of deterioration could be alleviated by pre-processing datasets with MBAF2.

Contemporary high-end CT scanners continue to push the physical boundaries of the available hardware, in terms of detector, tube, and gantry technology, in particular. The first purpose-built cardiac CT scanner (CardioGraphe, GE Healthcare, Chicago, IL, USA) was introduced in 2017 with the intention of addressing the growing need for a CT that is optimized for cardiac imaging in its performance and accessibility. Stereo CT technology consisting of two rapidly alternating sources of radiation along the Z-axis enables whole-heart imaging in a single heartbeat with a focused field of view. Combined with an ultrafast gantry rotation speed of 0.24 s per rotation, these novel technological advancements offer excellent image quality to a wide array of challenging patients [45]. While increasing rotation speed is advantageous for eliminating motion artifacts during coronary imaging, it also generates increased image noise, emphasizing the lingering prerequisite for denoising techniques. ASIR-CV is a vendor-specific iterative reconstruction technique that integrates the full statistical noise model of the CCTA image data and applies iterative comparisons of each captured projection to an incorporated projection integrating modeling of both system optics and system statistics for the reduction of image noise [37,39,46]. On the other hand, MBAF2—an algorithm developed specifically for this dedicated scanner—is a vendor-proprietary modified non-local means (NLM) algorithm intended to remove noise while maintaining small details, such as calcifications [25]. MBAF2 could be applied individually or combined with ASIR-CV resulting in a blended image with preserved sharpness of small details. During the reconstruction chain, the application of MBAF2 precedes ASIR-CV reconstructions, the input to the dedicated IR is hence less noisy. The relative proximity of CACS measurements to FBP with the combined utilization of MBAF2+ASIR-CV might be explained by this technical perspective.

The current study has a few limitations that need to be acknowledged. First of all, we have to note that although IRs are utilized in routine clinical practice, these newer reconstruction methods have originally been developed for contrast-enhanced CT imaging and MBAF2 has not been validated for CACS calculations to date. Moreover, no phantom studies are available for the comparison of these reconstruction algorithms to date. All images were acquired from a dedicated cardiac CT scanner and were reconstructed using the same workstation and software, which could affect the applicability of our results on scans obtained from different vendors. Finally, although we propose the potential utility of MBAF2 in radiation dose reduction, as of yet, we have not evaluated possible radiation dose-saving methods.

## 5. Conclusions

In the current investigation, we demonstrated that besides their denoising potential, both MBAF2+ASIR-CV and stand-alone ASIR-CV underestimates calcium volume and CACS compared to the gold-standard FBP algorithm which may result in the downgrade reclassification of individual CV risk. The rate of reclassification, however, was significantly lower using the combination of MBAF2 and ASIR-CV, as compared to ASIR-CV alone. With the steady increase in the number of performed cardiac CTs and the prevailing need for radiation-conscious approaches, contemporary IR methods should also be considered for clinical application in calcium scoring, their effect on CACS values, however, should be taken into account when evaluating CT datasets.

## Figures and Tables

**Figure 1 diagnostics-13-00789-f001:**
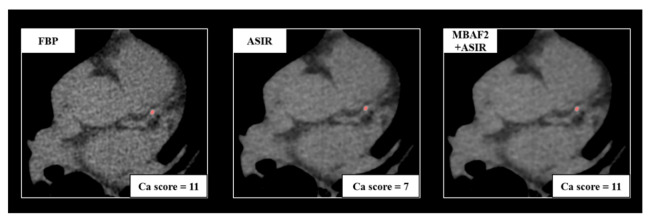
A representative example of a 61-year-old female patient with atypical chest pain. Acquisition acquired at routine dose reconstructed with FBP, ASIR-CV, and MBAF2+ASIR-CV. Both techniques markedly decreased image noise; however, the utilization of ASIR-CV resulted in the reclassification of the patient into a smaller cardiovascular risk group. FBP: filtered back projection; ASIR: adaptive statistical iterative reconstruction; MBAF2: model-based adaptive filter.

**Figure 2 diagnostics-13-00789-f002:**
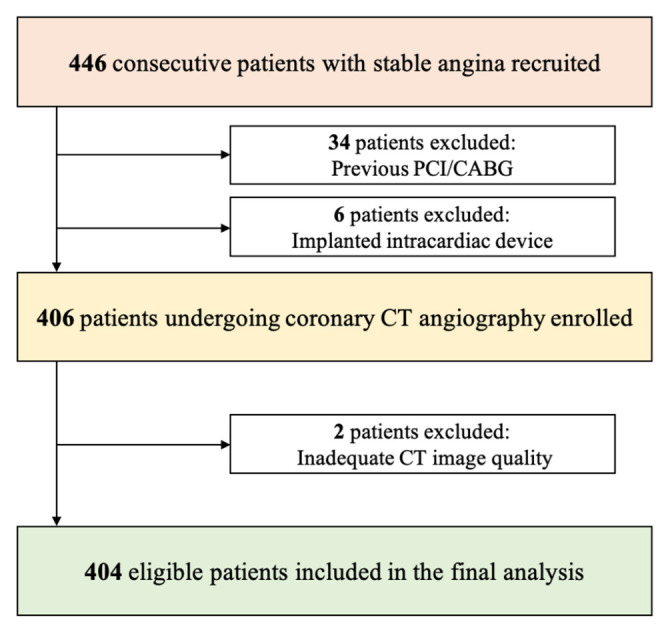
Flowchart of the study. PCI: percutaneous coronary intervention; CABG: coronary artery bypass grafting.

**Figure 3 diagnostics-13-00789-f003:**
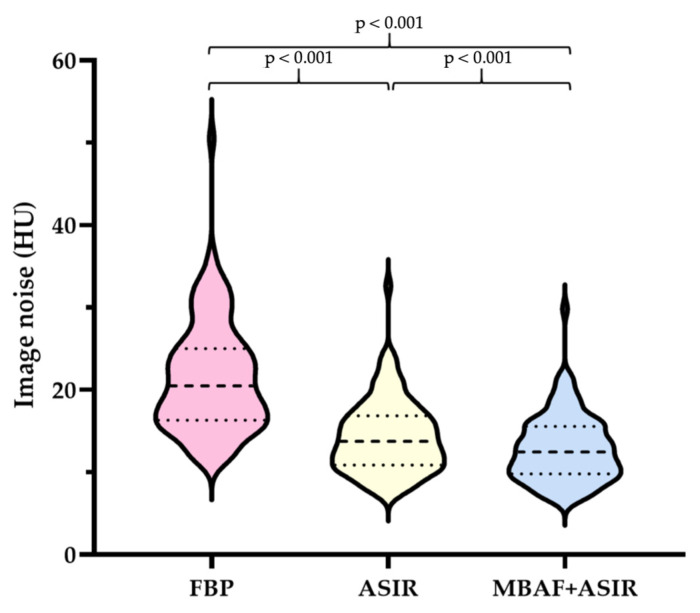
Violin plots showing the results of quantitative analysis of image noise obtained with FBP, ASIR-CV, and MBAF2+ASIR-CV algorithms. FBP: filtered back projection; ASIR: adaptive statistical iterative reconstruction; MBAF2: model-based adaptive filter.

**Figure 4 diagnostics-13-00789-f004:**
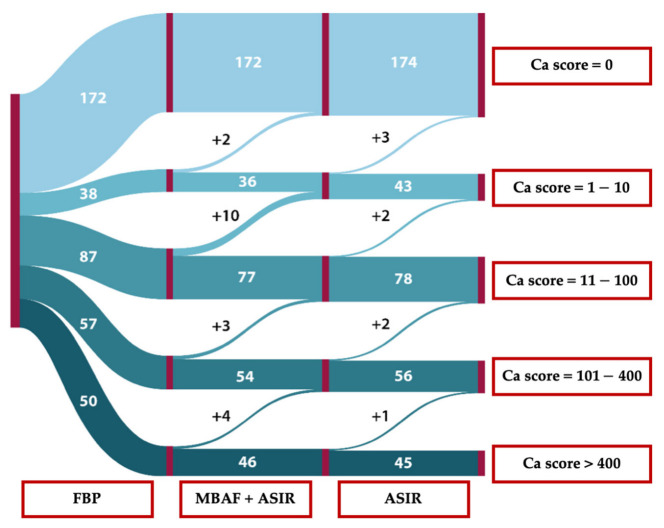
Sankey diagram of reclassification tendencies between different reconstruction methods. FBP: filtered back projection; MBAF2: model-based adaptive filter; ASIR: adaptive statistical iterative reconstruction.

**Figure 5 diagnostics-13-00789-f005:**
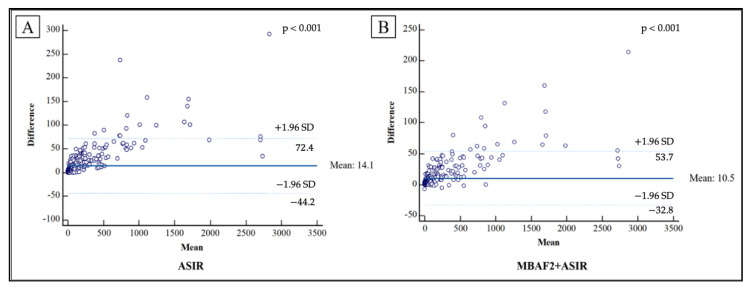
Bland–Altman plots demonstrating the differences observed between FBP and the different reconstruction techniques regarding Agatston scores. Compared with FBP, ASIR-CV underestimated calcium score by a mean of 14.1 (**A**), while on average, the combination of MBAF2 and ASIR-CV measured 10.5 less than FBP (**B**). FBP: filtered back projection; ASIR: adaptive statistical iterative reconstruction; MBAF2: model-based adaptive filter.

**Table 1 diagnostics-13-00789-t001:** Patient demographics and imaging parameters.

Parameters	n = 404
Demographics	
Age (years)	57.5 ± 12.2
Female sex, n (%)	146 (38.7)
BMI (kg/m^2^)	27.9 ± 4.6
Cardiovascular risk factors	
Current smoker, n (%)	64 (17.0)
Hypertension, n (%)	219 (78.1)
Diabetes mellitus, n (%)	51 (13.5)
Dyslipidemia, n (%)	135 (35.8)
CT characteristics	
DLP (mGy*cm)	17.7 ± 10.1
Effective dose (mSv)	0.3 ± 0.1
Average heart rate (1/min)	65.7 ± 15.1

Continuous variables are expressed as mean ± standard deviation (SD), while categorical variables are expressed as numbers and percentages. BMI: body mass index; CTA: CT angiography; DLP: dose length product.

**Table 2 diagnostics-13-00789-t002:** Overall calcium volume and Agatston scores acquired with different reconstruction strategies.

	FBP	ASIR-CV	MBAF2+ASIR-CV	*p*
Calcium volume (mm^3^)	7.0 (0.0–133.25)	4.0 (0.0–103.5)	5.0 (0.0–118.5)	<0.001
Agatston score	17.0 (0.0–324.8)	9.5 (0.0–286.5)	11.0 (0.0–288.0)	<0.001

Agatston score values and calcium volumes were compared with the Friedman test and post hoc comparisons were performed using Wilcoxon’s signed-rank test. All pairwise comparisons proved to be significant. FBP, filtered back projection; MBAF2, model-based adaptive filter; ASIR-CV, adaptive statistical iterative reconstruction.

**Table 3 diagnostics-13-00789-t003:** Concordance correlation coefficient of different reconstruction techniques with FBP.

	Calcium Volume (mm^3^)	Agatston Score
ASIR-CV	0.992	0.996
MBAF2+ASIR-CV	0.996	0.998

All pairwise comparisons have been performed versus the gold-standard FBP. FBP: filtered back projection; MBAF2: model-based adaptive filter; ASIR: adaptive statistical iterative reconstruction.

## Data Availability

The data presented in this study are available on request from the corresponding author. The data are not publicly available due to reasons pertaining to patient privacy.

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
