# Peer review of "The Impact of Novel Reconstruction Algorithms on Calcium Scoring: Results on a Dedicated Cardiac CT Scanner"

_diagnostics, 2023, doi:10.3390/diagnostics13040789_

Round 1

Reviewer 1 Report

The paper presents The Impact of Novel Reconstruction Algorithms on Calcium 2 Scoring: Results on a Dedicated Cardiac CT scanner.

Overall, the scientific objective is interesting and important.
The article is well written and comprehensive.
There are no ethical concerns about this study.
The references are up-to-date and the most important studies have been cited.
There are some revisions needed. Please provide a point-by-point response to the following queries.

1.       please explain the abbreviations used in the abstract

2.       Please provide a flowchart, which depicts the flow of information through the different phases enrolment of participants – how many were included and excluded, and the reasons for exclusions.

3.       Please provide a justification and statistical results based on your data for the conclusion in the abstract - The concomitant use   of ASIR-CV and MBAF2 may allow the reduction of noise levels, while maintaining similar CACS values as FBP measurements.

4.       In the article there is written that ‘Contemporary reconstruction algorithms yield the potential of reducing radiation exposure by denoising coronary CT angiography (CCTA) datasets’. Do you have any data on this problem in regard to presented algorithms?

5.       Did you verify if this results correlates with thee clinical cardiovascular risk or endpoints?

Reviewer 2 Report

In this manuscript, Vecsey-Nagy  et.al. compared the advanced adaptive statistical iterative reconstruction (ASIR-CV) and model-based adaptive filter (MBAF2) to the gold-standard filtered-back projection (FBP) calculations. They concluded that combination of ASIR-CV and MBAF2 may reduce the noise levels with similar CACS values as FBP measurements according to their result of CT images of 404 patients. Overall this is an interesting study, while the main conclusion of it was not clearly presented in the manuscript.
1. Any patients shifted from a lower risk group to a higher category?

2.Fig2, it might be better to add arrows on the shifting projection to make it more clear.

3.What is correlation method in table3, Spearman or Pearson?

4.It is better to perform a regression analysis instead of labeling a fixed value, according to the data distribution presented on the Fig3.

Reviewer 3 Report

This paper studied the reconstruction algorithm, focusing on the application in coronary artery disease. Precise early detection of cardiovascular disease is an important topic which has great practical values.

The methodology is sound and the results are promising.

The analysis on the related work is not adaquate. More recent work, especially in the image processing field, should be further reviewed and added to the reference.

The gender distribution in the subjects population is not balanced. Would gender factor influence the results?

The age range seems to be narrow, how can we generalize the conclusion to new age groups, such as for yonger people would the conclusion still hold?

The novelty of this paper should be more clearly explained.

Round 2

Reviewer 1 Report

The paper presents The Impact of Novel Reconstruction Algorithms on Calcium Scoring: Results on a Dedicated Cardiac CT scanner. Overall, the scientific objective is interesting and important. The article is well written and comprehensive. There are no ethical concerns about this study. The references are up-to-date and the most important studies have been cited. All recommended revisions were included into revised version of the manuscript and all queries were resolved. I do not have any further comments. It presents high scientific value and important to the field and in my opinion should be accepted for publication.

Reviewer 2 Report

The authors' responses and the revised manuscript have released my concerns, I have no further questions.